# Huntingtin loss in hepatocytes is associated with altered metabolism, adhesion, and liver zonation

Robert M Bragg[1], Sydney R Coffey[1], Jeffrey P Cantle[1], Shikai Hu[2,3], Sucha Singh[3], Samuel RW Legg[1], Cassandra A McHugh[1], Amreen Toor[1], Scott O Zeitlin[4], Seung Kwak[5], David Howland[5], Thomas F Vogt[5], Satdarshan P Monga[3,6,7], Jeffrey B Carroll[1,8]

Huntington's disease arises from a toxic gain of function in the *huntingtin* (*HTT*) gene. As a result, many HTT-lowering therapies are being pursued in clinical studies, including those that reduce HTT RNA and protein expression in the liver. To investigate potential impacts, we characterized molecular, cellular, and metabolic impacts of chronic HTT lowering in mouse hepatocytes. Lifelong hepatocyte HTT loss is associated with multiple physiological changes, including increased circulating bile acids, cholesterol and urea, hypoglycemia, and impaired adhesion. HTT loss causes a clear shift in the normal zonal patterns of liver gene expression, such that pericentral gene expression is reduced. These alterations in liver zonation in livers lacking HTT are observed at the transcriptional, histological, and plasma metabolite levels. We have extended these phenotypes physiologically with a metabolic challenge of acetaminophen, for which the HTT loss results in toxicity resistance. Our data reveal an unexpected role for HTT in regulating hepatic zonation, and we find that loss of HTT in hepatocytes mimics the phenotypes caused by impaired hepatic *β*-catenin function.

## Introduction

Huntington's disease (HD) is a fatal autosomal dominant neurodegenerative disease arising from the expansion of a glutamine-coding CAG tract near the 5′ end of *huntingtin* (*HTT*) (MacDonald et al, 1993). Progressive changes in mood, cognition, and characteristic movement symptoms have been mapped in people carrying HD mutations. Unfortunately, no disease-modifying treatments have been approved (Ross et al, 2014). Recently a major focus in the HD field has been *HTT* mRNA and protein (*HTT* and HTT)-lowering therapies, which have advanced to human clinical studies with a diverse range of *HTT*-lowering agents (Tabrizi et al, 2019). Notably, multiple companies are investigating small molecule splice modulator drugs that result in degradation of *Htt*'s mRNA via nonsense mediated decay—treatment with these drugs results in robust reductions of *Htt* in the liver, suggesting that better understanding the impact of HTT lowering on hepatic physiology is well justified (Bhattacharyya et al, 2021; Keller et al, 2022).

*HTT* is highly conserved and widely expressed (Saudou & Humbert, 2016). Population studies reveal that loss of function mutations in *HTT* are much rarer than expected by chance, suggesting *Htt* is poorly tolerant of these mutations (e.g., gnomAD—observed/expected pLoF in *HTT* = 0.12, pLI = 1) (Karczewski et al, 2020). Indeed, *Htt* knockout in mice results in early embryonic lethality (Duyao et al, 1995; Nasir et al, 1995; Zeitlin et al, 1995) and hypomorphic alleles that express less *Htt* than normal in humans cause a profound neurodevelopmental disorder whose symptoms are distinct from the HD (Rodan et al, 2014; Lopes et al, 2016). HTT has been associated with many cellular pathways and roles including regulating autophagy, transcription, vesicular trafficking, cellular polarity, adhesion, and others (Saudou & Humbert, 2016).

To better understand normal HTT functions, and how these might predict the risks for ongoing HTT-lowering studies in HD patients, we generated mice that lack *Htt* expression in hepatocytes. Mice lacking *Htt* in hepatocytes are viable and appear healthy, which enabled us to conduct studies on the impact of HTT loss in vivo. We observe many molecular, transcriptional, and physiological changes in mice which lack *Htt* expression in hepatocytes. Most notably, we observe a shift in the normal zonated pattern of gene expression in the liver, consistent with a loss of pericentral hepatocyte identity.

Hepatocyte identity and gene expression varies across several axes, including a gradient of regionalized gene expression changes between the pericentral and periportal vein regions

[1]Behavioral Neuroscience Program, Department of Psychology, Western Washington University, Bellingham, WA, USA   [2]School of Medicine, Tsinghua University, Beijing, China   [3]Division of Experimental Pathology, Department of Pathology, University of Pittsburgh School of Medicine, Pittsburgh, PA, USA   [4]Department of Neuroscience, University of Virginia, Charlottesville, VA, USA   [5]CHDI Foundation, Princeton, NJ, USA   [6]Pittsburgh Liver Research Center, University of Pittsburgh Medical Center and University of Pittsburgh School of Medicine, Pittsburgh, PA, USA   [7]Division of Gastroenterology, Hepatology and Nutrition, Department of Medicine, University of Pittsburgh School of Medicine, Pittsburgh, PA, USA   [8]Department of Neurology, University of Washington, Seattle, WA, USA

Correspondence: jeffcarr@uw.edu

(Russell & Monga, 2017). The acquisition and maintenance of pericentral gene expression in hepatocytes is under the positive control of β-catenin signaling, which in turn is activated by the paracrine release of specific Wnt species from liver endothelial cells (ECs) (Hu et al, 2022). In short, we observe transcriptional, histological, and physiological evidence of reduced pericentral hepatocytes in the livers of $Htt^{LKO/LKO}$ mice, suggesting a peri-portalization of the liver after HTT loss, very similar to what has been observed in mice with directly impaired hepatic β-catenin signaling (Tan et al, 2006; Yang et al, 2014; Preziosi et al, 2017; Hu et al, 2022).

# Results

### Generation and validation of hepatocyte-specific Htt knockout ($Htt^{LKO/LKO}$)

To generate mice lacking hepatic $Htt$, we crossed a previously described allele in which exon-1 of the $Htt$ is flanked by loxP sites ($Htt^{tm2Szi}$, hereafter $Htt^{fl/fl}$) (Dragatsis et al, 2000) to B6.Cg-Speer6-ps1$^{Tg(Alb-cre)21Mgn/}$J, yielding an allele we refer to as hepatocyte-specific $Htt$ knockout mice ($Htt^{LKO}$). $Htt^{LKO/LKO}$ mice were born in expected Mendelian ratios when crossing to homozygosity (Table S1), breed with normal fecundity as homozygotes (mean = 6.3, s.d. = 2.4 pups per litter) (Nagasawa et al, 1973) and are indistinguishable from littermates in terms of gross appearance and behavior. $Htt$ transcripts and protein are markedly reduced in the livers of $Htt^{LKO/LKO}$ mice (Fig 1A and B), whereas HTT expression is spared in a survey of other peripheral tissues based on Western blot analysis (Fig 1B and C). Complete HTT loss in bulk liver assessments is not predicted (or seen, Fig 1A–C), as our Cre line drives HTT knockout selectively in hepatocytes, but not the other cell types that comprise ~20% of liver cells (Poisson et al, 2017). These data suggest we successfully generated viable constitutive $Htt^{LKO/LKO}$ mice, which lack both $Htt$ and HTT in hepatocytes, but not in other organs examined.

### Physiological and adhesion phenotypes

We generated large cohorts of female $Htt^{+/+}$ and $Htt^{LKO/LKO}$ mice for metabolic analyses. Throughout their lives, $Htt^{LKO/LKO}$ mice weigh subtly less than $Htt^{+/+}$ mice (Fig 1D, linear mixed effects model genotype effect $F_{(1,30)}$ = 28.9, $P$ < 0.001). There are no corresponding differences in terminal organ weights (liver, perigonadal white fat, interscapular brown adipose tissue, spleen), when considered either as raw weights or normalized to body weights (Table S2). Plasma chemistry measurements in 6-mo-old $Htt^{LKO/LKO}$ mice (Table S3) reveal increased circulating blood urea nitrogen (Fig 2A; 29% increase; $N$ = 11/genotype, $t$ test $t_{(14.8)}$ = 3.8; $P$ = 0.002) and total cholesterol (Fig 2B; 35% increase; $N$ = 11/genotype, $t$ test $t_{(12.4)}$ = 3.6; $P$ = 0.003), whereas no differences were observed in the levels of liver enzymes alanine aminotransferase (Fig 2C, $N$ = 11/genotype, $t$ test $t_{(9.25)}$ = −0.2; $P$ = 0.88) or aspartate aminotransferase (Fig 2D, $N$ = 11/genotype, $t$ test $t_{(9.9)}$ = −0.21; $P$ = 0.84), indicating that these changes occur in the absence of gross liver damage in aged $Htt^{LKO/LKO}$ mice. We also found that $Htt^{LKO/LKO}$ mice have increased level of bile acids when tested at 2 mo (Fig 2E, 189% increase; $N$ = 163 ($Htt^{+/+}$) and 34 ($Htt^{LKO/LKO}$), $t$ test $t_{(40.5)}$ = 2.6; $P$ = 0.01). Collectively, these data reveal a pattern of altered systemic urea, cholesterol, and bile acid metabolism in $Htt^{LKO/LKO}$ mice.

Given the liver's key role in systemic glucose metabolism, and HTT's proposed controversial role in regulating organismal and cellular metabolisms (Brustovetsky, 2016), we investigated systemic glucose metabolism in $Htt^{LKO/LKO}$ mice 6 and 13 mo of age. At both ages, we observe significant 18-h fasting hypoglycemia in $Htt^{LKO/LKO}$ compared with $Htt^{+/+}$ mice (Fig 2F, ages pooled for plotting, 27% reduction; $N$ = 18 $Htt^{+/+}$, 22 $Htt^{LKO/LKO}$, $t$ test $t_{(29.9)}$ = 2.2; $P$ = 0.036). To probe changes in glucose disposal we conducted glucose-tolerance tests (GTT) and insulin-tolerance tests (ITT), finding no effect of genotype in plasma glucose levels during these challenges after normalization of baseline hypoglycemia (6-mo data shown in Fig 2G–I; GTT: $N$ = 10/genotype/age, ANOVA, $F_{(1,37)}$ = 1.1, $P$ = 0.3; Fig 2H ITT: $N$ = 10/genotype, ANOVA, $F_{(1,37)}$ = 7.26, $P$ = 0.88). Hepatic gluconeogenesis, as measured by the pyruvate tolerance test was also unchanged in $Htt^{LKO/LKO}$ mice compared with WT mice at 6 and 13 mo of age (Fig 2I; $N$ = 10/genotype, ANOVA, $F_{(1,37)}$ = 1.26, $P$ = 0.27). Together, these results suggest that whereas loss of HTT

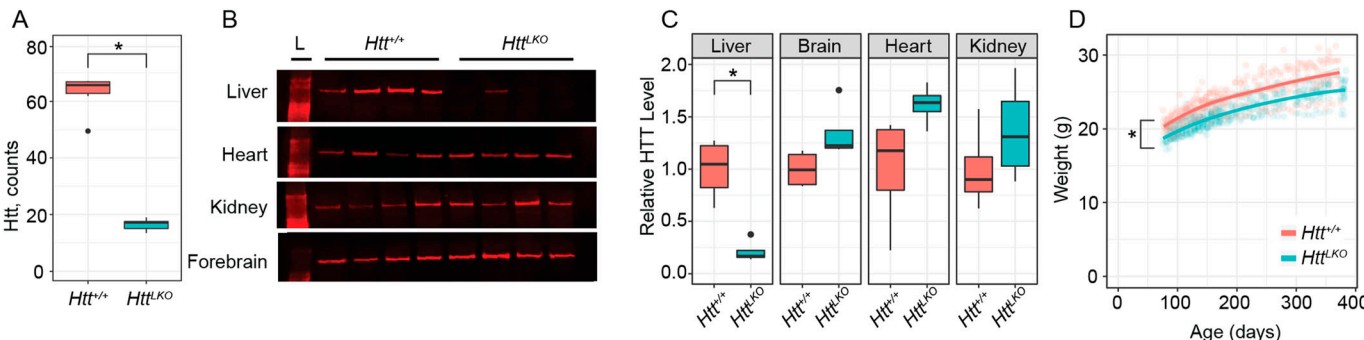

**Figure 1. Huntingtin (HTT) transcript, protein levels, and body weight of hepatocyte-specific HTT knockout mice ($Htt^{LKO/LKO}$).**
**(A)** RNAseq reveals $Htt$ mRNA is reduced in $Htt^{LKO/LKO}$ mice compared with WT littermates ($N$ = 5/genotype, $P$ < 0.0001). **(B, C)** Western blot reveals that liver HTT is reduced in $Htt^{LKO/LKO}$ mice compared with WT littermates ($N$ = 4/genotype, $t_{(3.8)}$ = 4.9; $P$ = 0.035), whereas the brain, heart, and kidney are unchanged (brain, $t_{(5.2)}$ = 2.2; $P$ = 0.33; heart, $t_{(3.7)}$ = 2.1; $P$ = 0.43; kidney, $t_{(5.8)}$ = 1.1; $P$ = 1). Bonferroni correction was used to adjust for multiple comparisons. **(D)** $Htt^{LKO/LKO}$ mice weigh less than WT littermates (9.5% reduction; $N$ = 11 WT, 20 LKO, $P$ < 0.0001). * indicates significant statistical difference.
Source data are available for this figure.

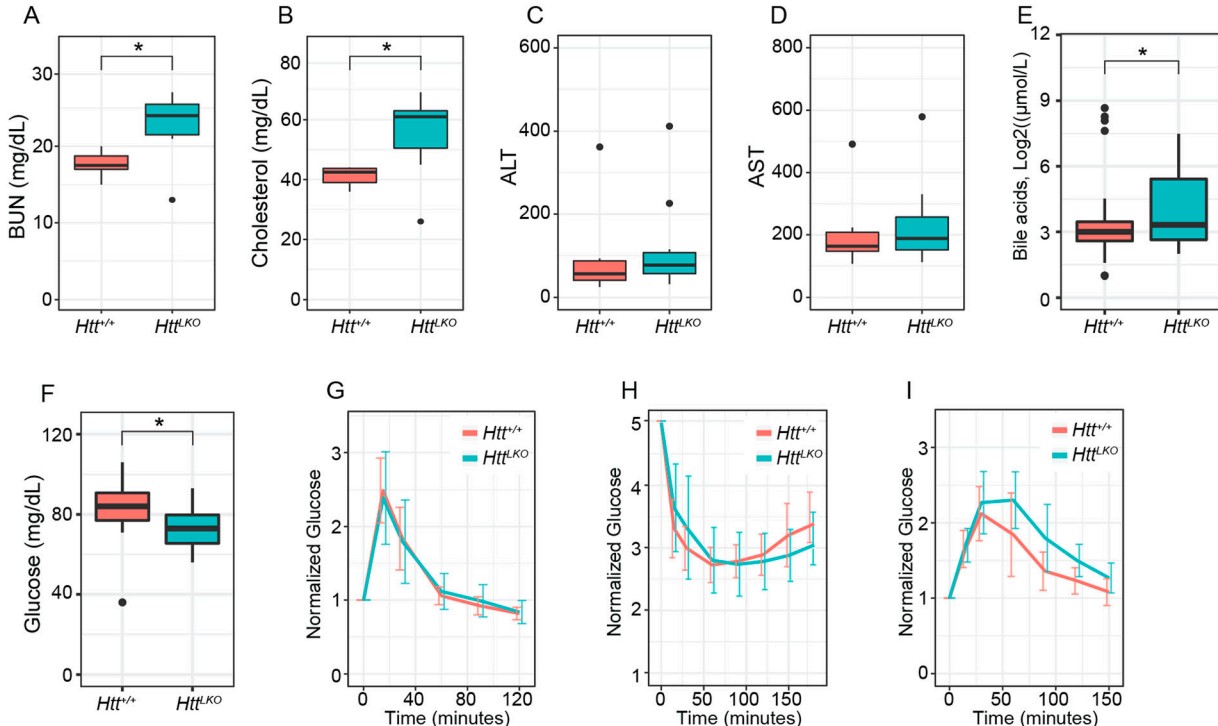

**Figure 2. Hepatocyte-specific *Htt* knockout mice have reduced circulating glucose, elevated urea, total cholesterol, and bile acids, but normal liver enzymes, glucose disposal, insulin sensitivity, and gluconeogenesis.**
**(A, B, C, D, E, F, G, H, I)** At 6 mo of age, *Htt^LKO/LKO^* mice have increased circulating blood urea nitrogen, (B) increased circulating total cholesterol, (C) normal ALT, and (D) AST (E) increased circulating bile acids and (F) reduced fasting glucose. *Htt^LKO/LKO^* mice display normal glucose clearance (G) when challenged with bolus of glucose in a glucose-tolerance test, display normal hypoglycemia, and recovery (H) when challenged with an insulin tolerance test, and no change in time required to convert pyruvate and clear the resulting glucose (I) in a pyruvate tolerance test. * indicates significant statistical difference.
Source data are available for this figure.

is associated with mild hypoglycemia, systemic glucose metabolism and gluconeogenesis are normal in *Htt^LKO/LKO^* mice.

During portal vein perfusions for the isolation of purified primary hepatocytes from *Htt^LKO/LKO^* mice, we observed abnormal blistering of Glisson's capsule, which surrounds the liver, because of the rapid collapse of the matrix of underlying hepatocytes (Fig 3A). This blistering of the liver capsule is observed in each *Htt^LKO/LKO^* mouse (N > 200 mice; Fig 3B). To test whether expanded polyglutamine HTT (mHTT) leads to similar adhesion changes, we examined liver blistering in *Htt^Q111/Q111^* mice, which express only mHTT, and observed no blistering (N = 20 mice). We also examined two different lines of mice with reduced HTT levels—first *Htt^LKO/+^* mice that express 50% WT HTT levels (N = 5 mice) and *Htt^Q175/Q175^* (N = 6), which are known to express only ~25% of HTT levels (Southwell et al, 2016). Neither line expressing low HTT levels produced any blistering, suggesting that HTT reduction to between 0% and 25% of WT levels is required for this phenotype to occur, and that it is solely caused by HTT loss, not mHTT expression.

### Transcriptional signature of *Htt^LKO/LKO^* livers reveals fate loss, altered zonation, and inflammation

To establish the transcriptional consequences of chronic HTT lowering in vivo, we conducted RNA sequencing (RNA-seq) from livers of 10-mo-old *Htt^LKO/LKO^* mice. We observe 730 genes that are significantly up-regulated and 389 genes that are down-regulated in the livers of 10-mo-old *Htt^LKO/LKO^* mice (log$_2$ fold-change (FC) cutoff of ±0.5, adjusted *P*-value cutoff of 0.05; Fig 4A). On manual inspection of a volcano plot (Fig 4A), a number of the most down-regulated genes were observed to be required for core hepatocyte functions. To more formally test this observation, we used the TissueEnrich algorithm (Jain & Tuteja, 2018), which uses publicly available cross-tissue gene expression data to determine whether a gene list has a higher than expected number of tissue-restricted genes. Interestingly, amongst 389 robustly *down*-regulated genes in *Htt^LKO/LKO^* livers, 81 are liver-restricted genes, a result which is highly unlikely to occur by chance (Fig 4B; *log$_2$(FC)* = 5.4, *P* = 2.8 × 10$^{-36}$). Of the 730 up-regulated genes, only 6 are liver-specific, not corresponding to an enrichment in liver-specific genes given a list of this size (Fig 1B; *log$_2$(FC)* = 0.25, *P* = 0.99), and a larger number are enriched for expression in other tissue types, including thymus and cortex (Fig 1B; *log$_2$(FC)* = 5.2, *P* = 1.6 × 10$^{-17}$; *log$_2$(FC)* = 2.6, *P* = 4.7 × 10$^{-7}$). Consistent with these gene-level changes, down-regulated genes of *Htt^LKO/LKO^* livers were enriched in gene-ontology pathways involving canonical hepatic functions (e.g., long-chain fatty acid metabolic processes - GO:0001676), whereas up-regulated pathways are less clearly related to hepatocyte functions, but notably feature a number of pro-inflammatory pathways, including "T cell activation—GO:0042110" and "cellular response to type I interferon—GO:0071357)" (Fig 4C and D).

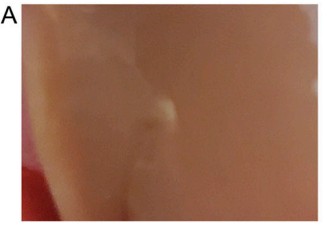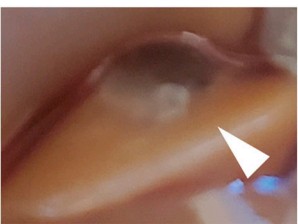

| Genotype | N | Estimated HTT | Blistering |
|---|---|---|---|
| WT | >200 | 100% | 0% |
| $HTT^{fl/fl}$ | 5 | 100% | 0% |
| $HTT^{Q111/+}$ | >200 | 100% | 0% |
| $HTT^{Q111/Q111}$ | 20 | 100% | 0% |
| $HTT^{Q175/Q175}$ | 6 | 25% | 0% |
| $HTT^{LKO/+}$ | 5 | 50% | 0% |
| $HTT^{LKO/LKO}$ | >200 | 0% | 100% |

**Figure 3. Hepatocyte-specific *Htt* knockout mice display impaired cellular adhesion.**
**(A)** WT liver (left) during perfusion shows normal intact liver and Glisson's capsule, whereas $Htt^{LKO/LKO}$ mice show swelling or "blistering" of the Glisson's capsule as it pulls away from the underlying hepatocyte matrix (right, arrowhead). **(B)** List of mice with modified *Htt* alleles and the corresponding percentage of mice that display blistering.

Concerned that these pro-inflammatory signals might result from immune cell infiltration, thereby changing the balance of cell types present in our bulk liver RNA-seq analysis, we scored five axes of potential hepatic pathology, including immune cell infiltration, hepatocyte swelling, fibrosis, lipid accumulation, and necrosis, finding no alterations in $Htt^{LKO/LKO}$ livers compared with $Htt^{+/+}$ (Table S4). We also profiled 42 inflammatory markers in the plasma of $Htt^{+/+}$ and $Htt^{LKO/LKO}$ mice at 6 and at 13 mo and found that 25 of 42 analytes provided results that were above the lower limit of detection and present in a robust number of animals (Table S5). Only very modest changes were observed in three circulating inflammatory markers, including an increase of tissue inhibitors of matrix metalloproteinases (TIMP-1; 2-way ANOVA, genotype effect $F_{(1, 21)}$ = 5.3; $P$ = 0.03) and modestly reduced levels of Eotaxin-1 in $Htt^{LKO/LKO}$ mice at both ages (two-way ANOVA, genotype effect $F_{(1, 21)}$ = 6.2; $P$ = 0.021). These results suggest that a frank increase in immune cell infiltration is unlikely to explain the pro-inflammatory gene expression we observe, with only limited changes observed in circulating inflammatory modulators.

*HTT* has been proposed to play a number of important roles in regulating transcription, in part via interactions with transcription factors (TFs) (Saudou & Humbert, 2016), so we examined which TF target genes are enriched in differentially expressed genes (DEGs) in $Htt^{LKO/LKO}$ livers. Using EnrichR to study enrichments amongst the ChIP Enrichment Analysis (ChEA 2022) dataset, we observe that genes which are up-regulated in $Htt^{LKO/LKO}$ livers are enriched amongst genes annotated as PRC2 complex member SUZ12-target

genes (62 of 1,019 genes, odds ratio = 1.77, adjusted $P$ = 0.02) (Lachmann et al, 2010; Kuleshov et al, 2016; Xie et al, 2021). In contrast, down-regulated DEGs in $Htt^{LKO/LKO}$ livers are enriched in genes regulated by canonical hepatic transcription factors, for example, LXR (81 of 1,578 genes, odds ratio = 3.2, adjusted $P$ < 1.3 × $10^{-13}$). Individual genes aberrantly up-regulated in $Htt^{LKO/LKO}$ livers include neurotransmitter receptors (e.g., *Adora1*, *Gria3*, adj $P$ = 0.015 and 0.003; Fig 5B) and a number of cell fate-determining transcription factor genes not normally expressed in adult hepatocytes, including *Runx2* and *Pax5* (adjusted $P$ = 0.024 and 0.03; Fig 5B). These data are consistent with a model in which loss of HTT in hepatocytes drives aberrant de-repression of important cell identity and fate-determining genes, including a notable up-regulation of PRC2-target genes.

### Liver zonation changes suggest impaired β-catenin signaling

Increased circulating urea in $Htt^{LKO/LKO}$ mice (Fig 2) suggests potential dysfunction of one of two major urea handling pathways–namely the urea cycle, which occurs in periportal (PP) hepatocytes, or glutamine synthesis, which is strictly localized to pericentral (PC) hepatocytes (Halpern et al, 2017). Expression of urea cycle genes in $Htt^{LKO/LKO}$ livers is overall unchanged (e.g., *Ass1*; Fig 6A; $t$ = −0.3, $P$ = 0.89), whereas levels of glutamine synthetase trend lower (*Glul*; Fig 6A, $log2FC$ = −0.71, $t$ = −2.66, $P$ = 0.1). In the adult mouse liver, *Glul* expression is restricted to a narrow band of PC hepatocytes (Halpern et al, 2017), reflecting the general zonation of hepatocyte functions along the PC/PP gradient. Cadherin proteins are also differentially expressed in these zones: PC regions express N-cadherin, whereas PP regions express E-cadherin (Doi et al, 2007). At the RNA level, we observe no change in Cdh2 (N-cadherin; Fig 7C) expression, whereas we see increases in Cdh1 (E-cadherin; Fig 7C; $log_2FC$ = 1.4, t = 7.5, adjusted $P$-value = 0.001). This was reflected histologically, where we observe a qualitatively altered expression pattern of E-cadherin immunoreactivity, but no change in N-cadherin immunoreactivity in the livers of 6-mo-old $Htt^{LKO/LKO}$ mice (Fig 7A). We also see a robust reduction in other PC-enriched proteins, including Cyp1a2 and Cyp2e1, both histologically (CYP1A2, 24% reduction, $t_{(17.9)}$ = 10.29, $P$ = 6.22 × $10^{-9}$; CYP2E1, 31% reduction, $t_{(8.6)}$ = 6.2, $P$ = 0.0002; Fig 7B) and transcriptionally (Cyp1a2, $t$ = −3.6, $P$ = 0.04; Cyp2e1, t = −4.1, $P$ = 0.02; Fig 7C). These findings are consistent with altered zonation in $Htt^{LKO/LKO}$ livers contributing to the observed gene expression alterations and metabolic findings, including impaired urea clearance.

To further probe the altered hepatocyte zonation in $Htt^{LKO/LKO}$ livers, we divided our list of DEGs into those up- and down-regulated in $Htt^{LKO/LKO}$ livers and compared those lists with a published consensus list of genes enriched in pericentral or periportal hepatocytes (Ghallab et al, 2019). Among the 34 DEGs ($P$ < 0.05) in $Htt^{LKO/LKO}$ livers annotated as "periportal", we observe no clear trend in the directionality of gene expression changes (Fig 6D, 12 genes up-regulated and 22 genes down-regulated in $Htt^{LKO/LKO}$ livers). In striking contrast, among the 95 DEGs in $Htt^{LKO/LKO}$ livers annotated as "pericentral", virtually all are down-regulated (Fig 6C, 5 genes are up-regulated and 90 genes are down-regulated in $Htt^{LKO/LKO}$ livers, $P$ < 0.05). Pericentral hepatocyte gene expression in development and adulthood is activated and maintained by Wnt/β-catenin signaling (Russell & Monga, 2017). Specific Wnts that are expressed in liver endothelial cells in the pericentral region, including Wnt2 and Wnt9b, control gene expression in

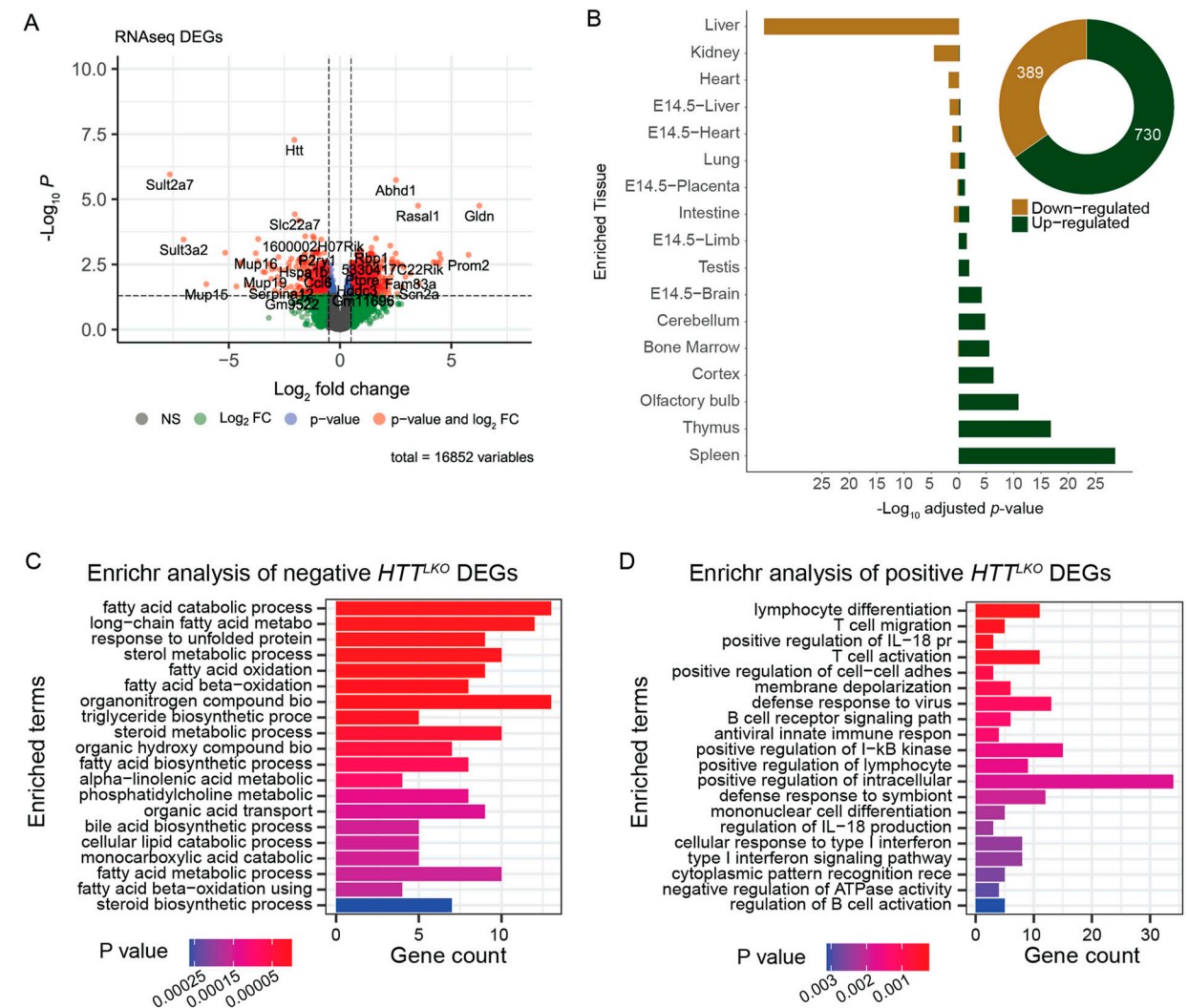

**Figure 4. Transcriptional analyses of livers from $Htt^{LKO/LKO}$ mice reveals down-regulation of hepatocyte-specific genes, and up-regulation of non-hepatic genes.**
**(A)** A volcano plot of 16,852 genes highlights 389 down-regulated ($Log_2$ FC < −0.5) and 730 up-regulated ($Log_2$ FC > 0.5) genes with adjusted $P$ < 0.05. **(B)** TissueEnrich shows that down-regulated genes are significantly overrepresented in genes annotated for expression in liver, whereas up-regulated genes are significantly overrepresented in non-liver tissue genes. **(C, D)** The top 20 enriched gene ontology pathways of biological processes ($P$ < 0.01) from down-regulated genes ($Log_2(FC)$ < −0.5 and adjusted $P$ < 0.05) are canonically liver-related pathways, whereas (D) the top 20 enriched gene ontology pathways of biological processes ($P$ < 0.01) from up-regulated genes ($Log_2(FC)$ > 0.5 and adjusted $P$ < 0.05) are not hepatocyte-specific pathways.
Source data are available for this figure.

hepatocytes at this region (Wang et al, 2015; Hu et al, 2022). The expression of *Wnt2* and *Wnt9b* is down-regulated in $Htt^{LKO/LKO}$ livers (Fig 6B; *Wnt2*: $Log_2(FC)$ = −1.2, $t$ = −3.7, $P$ = 0.037; *Wnt9b*: $Log_2(FC)$ = −1.29, $t$ = −5.5, $P$ = 0.006), consistent with the pericentral phenotype observed in $Htt^{LKO/LKO}$, suggesting a non-cell autonomous link between hepatocyte *Htt* expression and endothelial *Wnt* expression. Thus, convergent histological and transcriptional data suggest a loss of pericentral hepatocyte identity in $Htt^{LKO/LKO}$ livers, which is consistent with impaired $\beta$-catenin signaling after *Htt* loss.

### Acetaminophen (APAP) challenge

Mice with impaired hepatic $\beta$-catenin signaling experience "periportalization" of the liver, similar to our observations in $Htt^{LKO/LKO}$ livers. This change results in differential sensitivity to a range of challenges, including iron overload (Preziosi et al, 2017), ethanol intoxication (Liu et al, 2012), and APAP overdose (Apte et al, 2009). In the liver, APAP is metabolized by CYP2E1 and CYP1A2 into N-acetyl-p-benzoquinone imine, which is in turn toxic to hepatocytes (Raucy et al, 1989). Both CYPs are $\beta$-catenin target genes expressed pericentrally, rendering mice with hepatocyte-specific deletion of $\beta$-catenin resistant to APAP toxicity via lower levels of these enzymes (Yang et al, 2014). Consistent with our hypothesis, we observe that $Htt^{LKO/LKO}$ mice have markedly reduced circulating liver enzymes ALT (79.6%, $t_{(13.7)}$ = 3.4, $P$ = 0.004; Fig 8A) and AST (80% reduction, $t_{(12.7)}$ = 3.6, $P$ = 0.003; Fig 8B) after a 20-h APAP challenge. This confirms that $Htt^{LKO/LKO}$ mice share physiological sensitivities to those of mice with impaired $\beta$-catenin signaling and

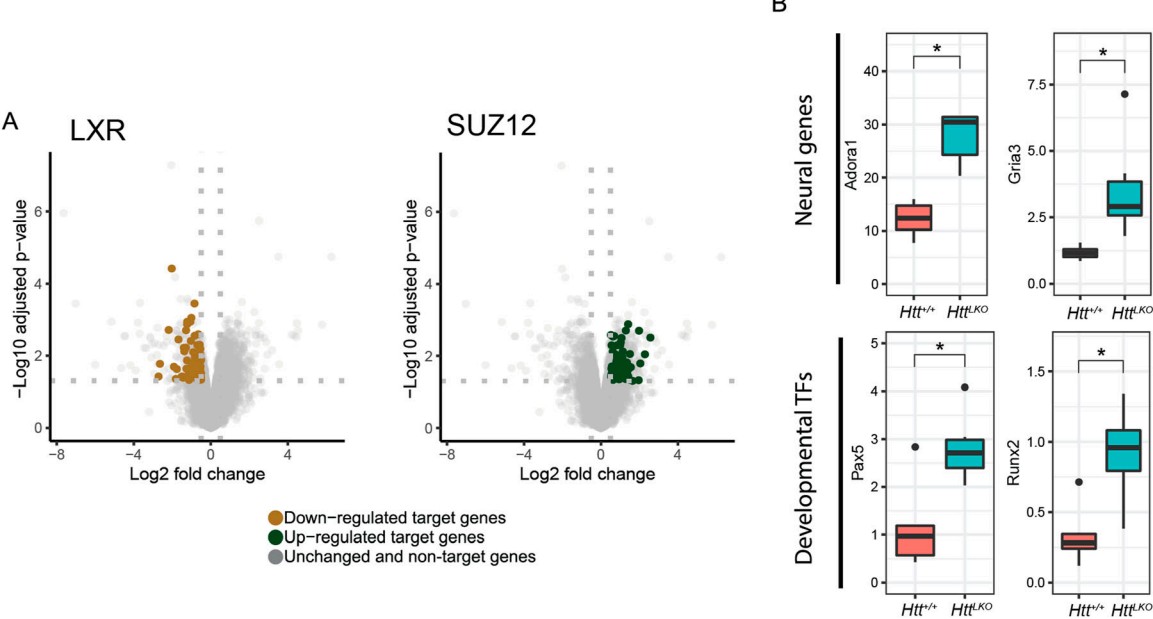

**Figure 5. Transcriptional analyses of livers from HttLKO/LKO mice reveal the down-regulation of hepatocytes-specific genes, and up-regulation of non-hepatic genes.**
**(A)** Down-regulated differentially expressed genes in $Htt^{LKO/LKO}$ livers are enriched with genes regulated by canonical hepatic transcription factors, including LXR, whereas up-regulated differentially expressed genes are enriched with genes regulated by the PRC2 complex SUZ12–target genes. **(B)** Neural genes *Gria3* and *Adora1* are up-regulated in $Htt^{LKO/LKO}$ livers, as are developmental transcription factor genes *Pax5* and *Runx2*. * indicates significant statistical difference.

further supports liver periportalization in mice lacking HTT in hepatocytes.

## Discussion

Despite more than 30 yr of intensive studying after the identification of the causal gene in Huntington's disease, there remains much to learn about HTT function—especially outside of the central nervous system. We developed and characterized mice in which *Htt* is excised from hepatocytes, leading to important pathophysiological alterations owing to observed transcriptional changes. Our work examining the consequences of HTT knock-out in hepatocytes expands the understanding of WT HTT functions, and describing an unexpected role in cellular fate and liver zonation.

A primary motivation for this study is to better understand loss of function liabilities in on-going HTT-lowering trials in HD patients (Tabrizi et al, 2019). One approach to HTT lowering in clinical studies in HD patients involves small-molecule RNA-splicing modulators which induce nonsense-mediated decay of *HTT* transcript, reducing both WT and mutant HTT proteins (Bhattacharyya et al, 2021; Keller et al, 2022). This approach is expected to reduce HTT levels in central and peripheral tissues, notably including the liver, which is exposed to high levels of oral drugs (Almazroo et al, 2017). Our data suggest that these trials should include monitoring for changes seen in $Htt^{LKO/LKO}$ mice including hypoglycemia, hypercholesterolemia, uremia, and increased circulating bile acids (Fig 2). Our APAP challenge results provide functional evidence that HTT loss in hepatocytes impacts clinically relevant hepatic physiology,

including the first pass metabolism of drugs by Cyp enzymes (Fig 7). More broadly, hepatic dysregulation of β-catenin signaling and consequential alterations in zonation are associated with important physiological consequences. Missense mutations in Wnt co-receptor LRP6 lead to early coronary artery disease in human mutation carriers (Mani et al, 2007) and knockin mice, who develop hepatosteatosis, even on regular chow diets (Go et al, 2014). Many supportive mouse studies have established that impaired β-catenin signaling leads to liver periportalization which is associated with increased vulnerability to ethanol intoxication (Liu et al, 2012), iron overload (Preziosi et al, 2017), and resistance to APAP challenge (Yang et al, 2014). Although the expression of several periportal genes is not altered in the absence of hepatic HTT, it is likely that their distribution across the liver lobule may be altered and thus the observation of periportalization of the liver is in fact true and functional. We have recently shown that loss of *Wnt2* and *Wnt9b* together from the endothelial cells did not alter the overall expression of periportal genes like *Ass1*, but instead, led to their appearance in pericentral zone hepatocytes (Hu et al, 2022). This is planned for future studies using single-cell spatial transcriptomics.

Decreased Wnt-β-catenin signaling in the $Htt^{LKO/LKO}$ mice is also supported by additional observations. Notably, mice with β-catenin knockout in hepatocytes show increased basal bile acid levels similar to $Htt^{LKO/LKO}$ (Behari et al, 2010). These mice also exhibit a basal compensatory increase in farnesoid X receptor activation, leading to decreased bile acid synthesis from cholesterol and higher basal cholesterol levels, again similar to $Htt^{LKO/LKO}$ mice. Finally, mice with hepatic β-catenin knockout showed higher basal

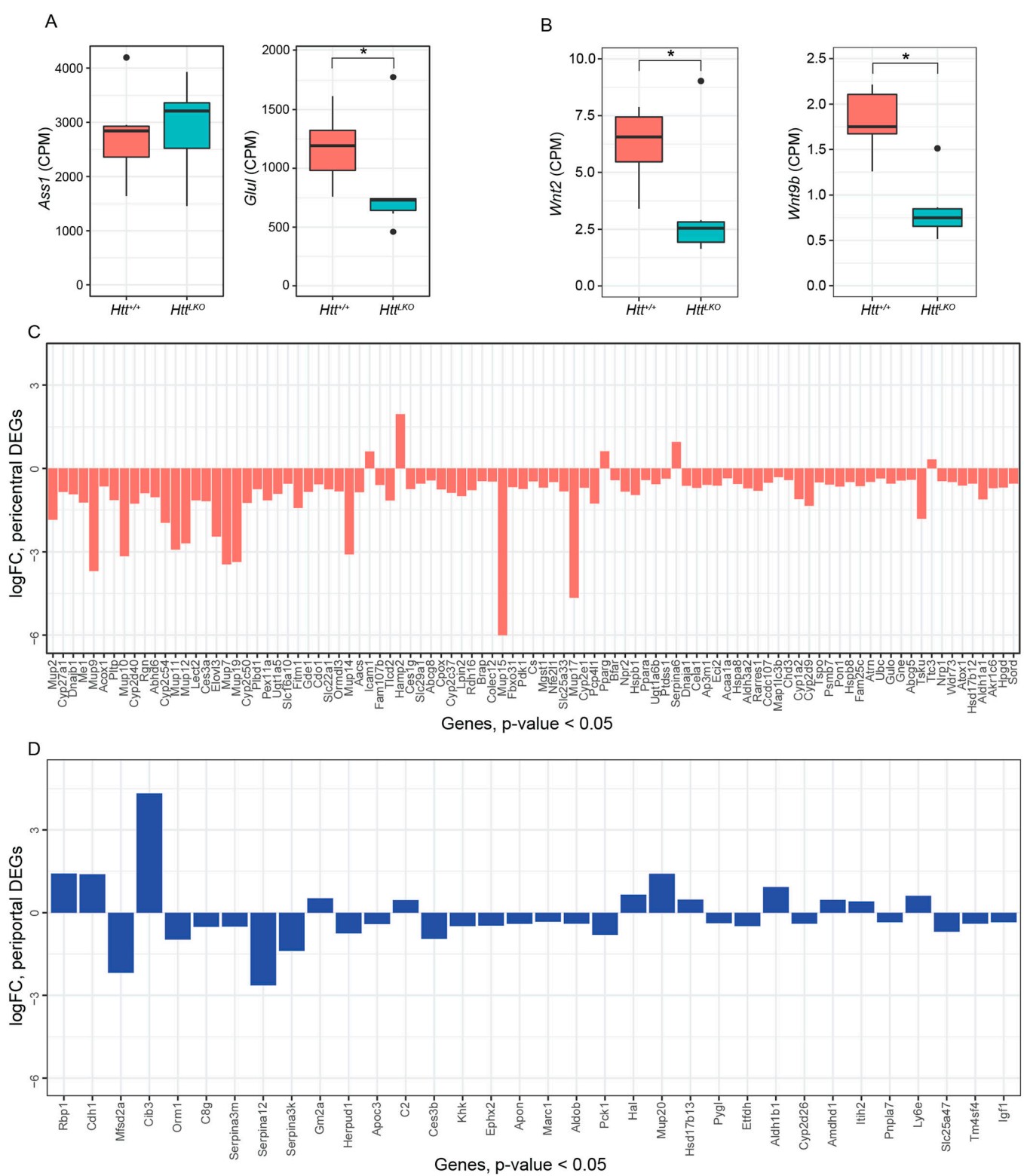

**Figure 6. Hepatic transcriptional analyses from *Htt*^LKO/LKO^ mice reveal the down-regulation of pericentral genes.**
**(A)** Argininosuccinate synthase 1 (*Ass1*) expression is unchanged, whereas glutamine synthase (*Glul*) is modestly decreased. **(B)** The transcripts of specific Wnt genes are down-regulated, *Wnt2* and *Wnt9b*. **(C, D)** LKO differentially expressed genes (Log$_2$(FC) < −0.5 or Log$_2$(FC) > 0.5 and *P* < 0.05) that are annotated as pericentral genes are near-exclusively negative, whereas (D) differentially expressed genes that are manually annotated as periportal do not follow a predominant pattern.* indicates significant statistical difference.

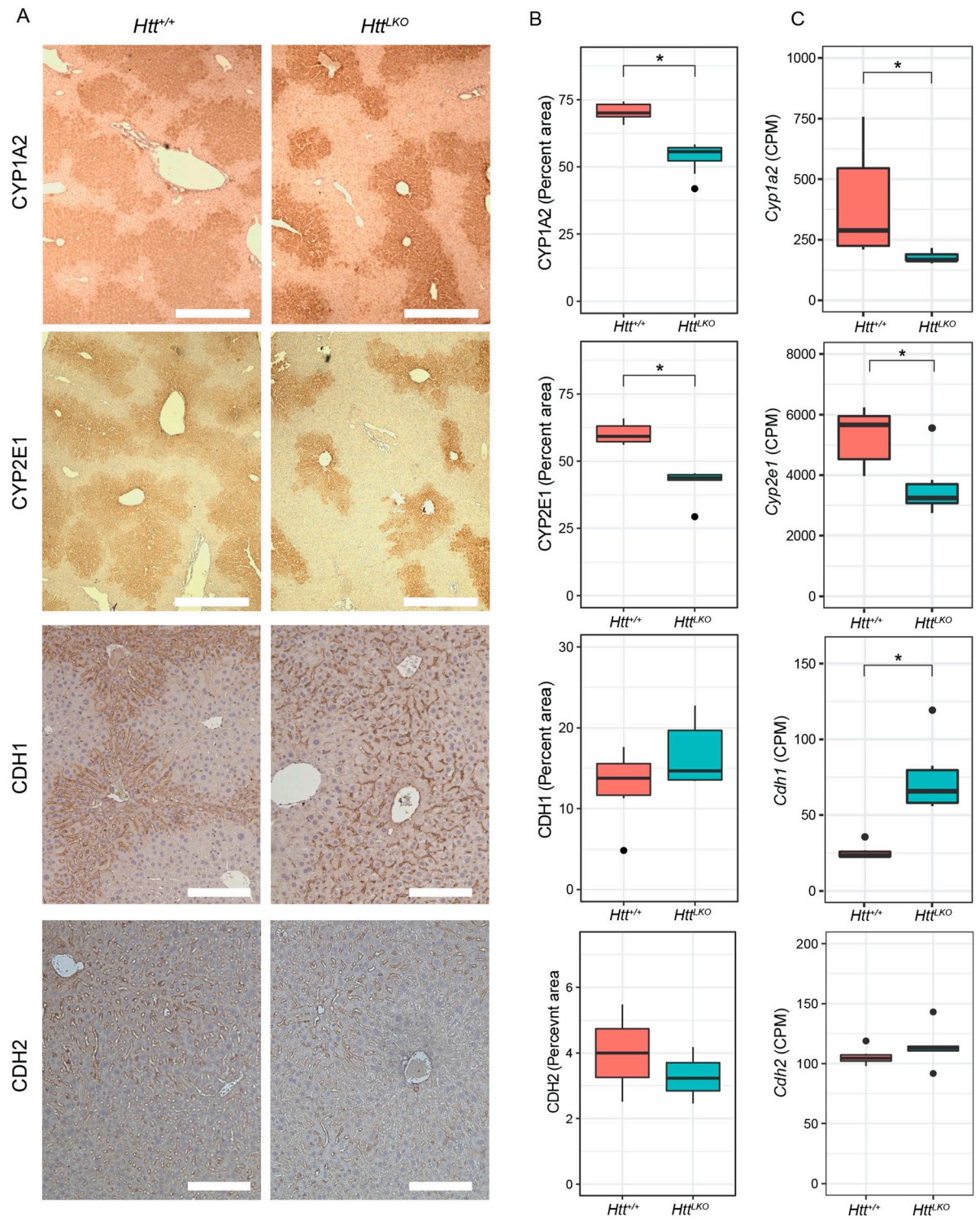

**Figure 7. The livers of mice with *huntingtin*-deficient hepatocytes have altered zonation.**
**(A, B, C)** Expression patterns of pericentral markers CYP1A2 and CYP2E1, but not CDH2, show reduced expression from pericentral zone into the periportal zone, whereas periportal marker CDH1 shows a qualitatively increased expression near the pericentral zones, yet nonsignificant alteration in total expression. Quantified IHC (B) and RNA sequencing (C) show reduced expression of pericentral proteins in *Htt*^{LKO/LKO} mice (Cyp1a2, Cyp2e1) and corresponding transcripts (RNAseq), whereas periportal marker CHD1 shows unchanged IHC (*P* = 0.17) but increased transcript levels *P* = 0.001. Scale bar = 350 *µ*m (CYP1A2, CYP2E1) or 100 *µ*m (CDH1, CDH2).* indicates significant statistical difference.
Source data are available for this figure.

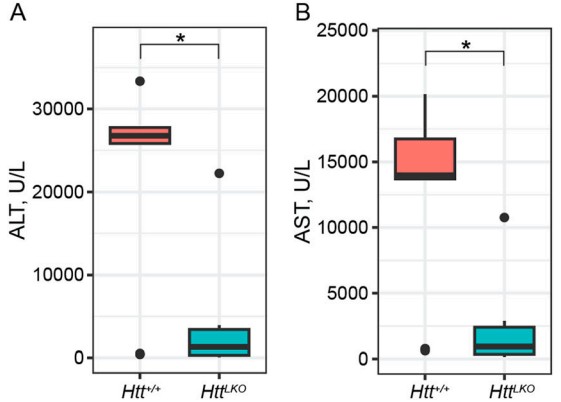

**Figure 8. Htt^LKO/LKO mice are resistant to a toxic acetaminophen challenge.**
**(A, B)** Circulating levels of hepatic enzymes ALT (A) and AST (B) were quantified after a 20-h acetaminophen challenge. This reveals a markedly lower level in Htt^LKO/LKO mice compared with WTs (79.6% reduction of; 80% reduction of AST; Htt^+/+ N = 9, Htt^LKO/LKO N = 7). * indicates significant statistical difference. Source data are available for this figure.

NF-κB activation and increased baseline inflammation (Nejak-Bowen et al, 2013). Indeed, Htt^LKO/LKO mice also display a higher T cell signature in our RNA-seq analysis and modest changes in circulating pro-inflammatory modulators, although the exact mechanism of this observation requires further investigation.

Whereas only a limited number of studies have investigated the role of WT HTT on metabolism, a broader literature exists on the impact of polyglutamine-expanded mHTT expression on cellular and organismal metabolisms. Urea cycle dysfunction has been described in both HD patients and HD model mice, which is improved in mouse models via a low-protein diet, leading to improvements in HD-relevant symptoms in parallel to improved urea cycle function (Chiang et al, 2007). These urea cycle deficiencies were proposed to be rooted in reduced transcription of key urea cycle genes, including argininosuccinate lyase and argininosuccinate synthetase by the C/EBPα transcription factor. Subsequently, markedly increased urea was observed in the brains of both HD transgenic model sheep and HD patients (Patassini et al, 2015; Handley et al, 2017). Although there is no evidence of clinically impactful hepatic dysfunction in HD patients, an elegant series of studies probed hepatic mitochondrial function in HD mutation carriers using ingestion of $^{13}$C-labeled methionine, which is catabolized in hepatic mitochondria to $^{13}CO_2$ that can be monitored in the breath (Stuwe et al, 2013; Hoffmann et al, 2014). These studies revealed early, progressive reductions in $^{13}CO_2$ release in HD mutation carriers, consistent with the reduced number or function of hepatic mitochondria.

Amongst many other cellular pathways, HTT plays roles in transcriptional regulation. HTT can bind and regulate many transcription factors and transcriptional regulatory proteins, including key members of the basal transcriptional machinery (Steffan et al, 2000; Zhai et al, 2005; Seong et al, 2009; Gao et al, 2019), and regulate cytoplasmic retention of transcriptional regulators including repressor element 1-silencing (Zuccato et al, 2003; Shimojo, 2008). Consistent with HTT's roles in transcriptional regulation, we observed that loss of Htt is associated with a wide range of transcriptional changes in the livers of Htt^LKO/LKO

mice (Fig 4). Several notable observations emerge from our transcriptional data. First, the expression of pericentral genes are markedly reduced, whereas periportal genes show no clear direction of change (Fig 6C and D). Second, genes in core pathways associated with liver function, which are the targets of hepatic transcription factors such as LXR, are down-regulated (Fig 5). Third, non-hepatocyte genes, and PRC2–complex-target genes, are up-regulated (Fig 5). These data provide additional confirmation about the periportalization of the liver, and suggest that an up-regulation of cell-type inappropriate genes is a molecular phenotype associated with Htt loss.

We have previously observed that the expression of mHTT in the striatum of Htt^Q111/+ mice is associated with an unexpected scrambling of the gene expression profiles across a wide range of cell types: in every cell type, cell type-appropriate genes are down-regulated, whereas inappropriate genes are up-regulated (Malaiya et al, 2021). Other groups had previously observed alterations in tissue-specific gene expression in HD tissues (Achour et al, 2015; Langfelder et al, 2016; Le Gras et al, 2017), suggesting that there may be a convergence of gain- and loss-of-function phenotypes for mHTT, in that both appear to be driving a loss of cell fate commitment/homeostasis. Although β-catenin-mediated transcriptional regulation is a widespread pathway, the β-catenin protein itself lacks both a transactivation domain and DNA-binding domain, and is expressed across many different cell types and developmental stages (Söderholm & Cantù, 2021). This conundrum has led to the suggestion that β-catenin's role in transcriptional regulation plays a highly tissue- and developmental stage-dependent role in crafting the transcription via dynamic binding to partner proteins—first in the cytoplasm and then in the nucleus—to carry out cell- and developmental-stage selective tuning of gene expression.

Previous work has demonstrated functional interactions between HTT and β-catenin, where HTT levels have been shown to be regulated via direct interactions with multiple members of the destruction complex (Godin et al, 2010). This interaction has been proposed to aberrantly stabilize β-catenin, leading to enhanced mHTT toxicity in neurons, which was subsequently confirmed in a Drosophila model of HD, in which reduced Wingless/Wnt signaling was protective and overexpression of armadillo/β-catenin was deleterious (Dupont et al, 2012). More recently, investigation into neurodevelopmental functions discovered that HTT plays a role in RAB11-mediated N-cadherin trafficking during neuronal polarization (Barnat et al, 2017), and that HTT co-localizes with junctional complex proteins, including β-catenin, in the ventricular zone during neurogenesis in human embryos (Barnat et al, 2020). This latter study suggests that mHTT expression in the developing brain is associated with perturbations in β-catenin-mediated functions, consistent with our finding that loss of WT HTT perturbs liver zonation.

A limitation of our work in extending to HTT-lowering agents in clinical trials in HD patients is that our mouse lacks HTT expression both during development and through adulthood, as the Cre driver we have used is active as early as E10.5 (Weisend et al, 2009). This is very different from a human HD patient exposed to HTT lowering in the liver after being treated with agents that are being tested in adult humans. We are conducting ongoing investigations, which

focus on HTT knockout and knockdown in adult mice, to determine the extent to which the phenotypes we describe here are developmental in origin.

In sum, our findings implicate HTT in the regulation of hepatocyte cell fate, liver zonation, and liver physiology via regulation of β-catenin signaling and identify an intriguing axis of HTT–Wnt–β-catenin crosstalk requiring further mechanistic work. Our studies also have important and unexpected implications for HTT-lowering treatments that reach the liver.

# Materials and Methods

## Mice

We generated a hepatocyte-specific C57BL/6 *Htt* knockout mouse (*Htt*$^{LKO}$) by first back-crossing *Htt*$^{tm2Szi}$ (*Htt*$^{fl/fl}$ [Dragatsis et al, 2000]) to a C57BL/6 background using speed congenics (Charles River) and confirmed to be 99.9% B6 based on a 384 SNP panel and then crossing to B6.*Cg-Tg(Alb-cre)*$^{21Mgn/J}$ (stock 003574; JAX) (Postic et al, 1999). All *Htt*$^{LKO}$, *Htt*$^{fl/fl}$, and *Htt*$^{+/+}$ mice were bred at Western Washington University and housed in cages of 3–6 mice with access to food and water ad libitum unless otherwise mentioned. Vivarium lights were on a 12-h light/dark cycle. The Western Washington University Institutional Animal Care and Use Committee approved the generation of the *Htt*$^{LKO}$ mice and all procedures under protocols 16-006, 14-006, and 23-70.

## Immunoblotting

Total HTT levels were quantified via Western blotting. Tissues were homogenized in tubes with 1.5-mm zirconium oxide beads for 2 min at 6 m/s using a benchtop homogenizer (Beadblaster) in RIPA buffer (150 mM NaCl, 25 mM Tris–HCl, 1% NP-40, 1% sodium deoxycholate, 0.1% SDS) containing Halt protease and phosphatase inhibitors (Thermo Fisher Scientific). Concentration of protein lysates was determined by BCA assay (Pierce) according to the manufacturer's protocol.

Equal amounts of reduced protein were loaded into 3–8% Tris–Acetate Mini Gels (Thermo Fisher Scientific) and separated electrophoretically by molecular weight. Protein was transferred at 4°C for 17–20 h to a PVDF membrane. Membranes were stained with a REVERT total protein stain (LiCor) according to the manufacturer's protocol and imaged for 2 min at 700 nm on an Odyssey Fc (LiCor) to quantify the total protein in each lane for downstream normalization. Total protein dye was removed and membranes were blocked (Odyssey blocking buffer; LiCor) for 1 h at RT and probed overnight at 4°C with a monoclonal rabbit anti-HTT antibody (EPR5526; Ab_109115; Abcam) diluted 1:1,000 in blocking buffer plus 0.2% tween, followed by a 1 h RT incubation in goat anti-rabbit IgG IRDye 680RD-conjugated secondary antibody (926-68071; Ab_10956166; LiCor) diluted 1:15,000 in blocking buffer plus 0.2% tween and 0.01% SDS. Membranes were imaged for 2 min at 700 nm on an Odyssey Fc and quantified in Image Studio (LiCor). HTT protein was quantified by normalizing the HTT band signal to the total protein signal in each lane.

## RNA sequencing

The liver tissue for RNA sequencing (RNA-seq) was harvested from female *Htt*$^{+/+}$ and *Htt*$^{LKO/LKO}$ mice (N = 6 per genotype) and euthanized at 289 (SD = 13.7) days old. RNA was extracted (RNeasy Lipid Tissue Mini Kit; Qiagen) and confirmed for RIN > 7 (2100 Bioanalyzer; Agilent). cDNA libraries were constructed at Azenta using the Illumina TruSeq RNA Sample Prep Kit using and sequenced on a HiSeq 2000 (2 × 150 bp) to a read depth of $7.6 × 10^7 ± 9.9 × 10^6$. Sequence reads were trimmed to remove adapter sequences and nucleotides with poor quality using Trimmomatic v.0.36. Reads were aligned to the *Mus musculus* GRCm38 reference genome assembly (ENSEMBL) using STAR aligner v.2.5.2b. Unique gene hit counts were calculated by using featureCounts from the Subread package v.1.5.2. Differential gene expression was conducted using the edgeR (Robinson et al, 2010), with voom (Law et al, 2014) and pathway enrichment was assessed using enrichR which calculates and "odds ratio" to determine if DEGs are overrepresented for a given gene set (Chen et al, 2013; Kuleshov et al, 2016).

## Metabolic challenges

For the pyruvate tolerance test, mice were fasted at the beginning of their dark phase (6 pm) for 18 h and IP-injected with pyruvate (2,000 mg/kg) at 12 pm on Monday. Blood glucose from tail prick was measured (Contour next EZ glucometer; Bayer) at 0, 15, 30, 60, 90, 120, and 150 min post injection. For the GTT, mice were fasted for 6 h at 8 am on Wednesday and IP-injected with glucose (2,000 mg/kg) at 2 pm. Blood glucose was measured as above at 0, 15, 30, 60, 90, and 120 min post-injection. For the ITT, mice were fasted for 6 h on Friday at 8 am before IP insulin injection (0.75 U/kg; Henry Schein Animal Health). Blood glucose was measured as above at 0, 15, 30, 60, 90, 120, 150, and 180 min post-injection.

These tests began at either 160 (±2) or 366 (±6) d of age. *Htt*$^{+/+}$ mice with high bile acids (>20 µmol/liter) were excluded from metabolic testing as this correlates to high likelihood of portosytemic shunts (Cudalbu et al, 2013). All mice received an intravenous tail vein injection of 150 µl sterile saline 3 mo before testing. *Htt*$^{+/+}$ mice were imported from Jackson Labs 3.5 mo before testing.

For the APAP challenge, 94 ± 7-d-old mice were IP-injected with 300 mg/kg APAP (Sigma-Aldrich) freshly dissolved in PBS at 3 pm and were euthanized 20 h later. Cardiac blood was processed and measured for ALT and AST as described below.

## Blistering assay

To probe mice for adhesion deficits, the mice were perfused via the hepatic portal vein with Hanks' buffered saline solution (Gibco) without calcium and magnesium with EGTA (0.5 mM). Upon cannulation of the hepatic portal vein, the inferior vena cava (IVC) was cut and perfusion began immediately at ~10 ml/min for 5 min. Intermittent clamping (5 s, twice per minute) of the IVC was done to ensure that the buffer completely filled all liver lobes. Care was taken to ensure that IVC clamping would not burst the Glisson's capsule in livers that displayed blistering. *Htt*$^{Q175/Q175}$ and *Htt*$^{Q111/Q111}$ mice were imported from Jackson Labs 2 wk before testing.

## Histology

Left lateral and caudate liver lobes were collected from 6-mo-old (165 ± 9) male mice and drop-fixed in 10% neutral buffered formalin (BBC biochemical) for 72 h. After fixation, livers were stored in PBS + 0.02% sodium azide until they were paraffin embedded, cut into 5-$\mu$m sections, and mounted on glass slides.

For immunohistochemistry, sections were deparaffinized and rehydrated to distilled water as described previously (Bragg et al, 2017). Sections underwent heat-mediated antigen retrieval in a pressure cooker for 20 min in pH = 6 sodium citrate buffer (CYP2E1, CYP1A2). After cooling for 30 min, samples were placed in 3% $H_2O_2$ for 10 min to quench endogenous peroxide activity. After washing with PBS, slides were blocked for 10 min with Super Block (ScyTek). The primary antibodies were incubated at the following concentrations in PBS: CYP2E1 (HPA009128, 1:100; Sigma-Aldrich), CYP1A2 (sc53241, 1:100; Santa Cruz Biotechnology) N-cadherin (ab18203, 1:100; Abcam), E-cadherin (14472, 1:100; Cell Signaling). Samples were washed with PBS three times and incubated with the appropriate biotinylated secondary antibody (Vector Laboratories) diluted 1:250 (CYP2E1, CYP1A2) for 15 min at RT in antibody diluent or 1:500 (N-cadherin, E-cadherin) for 30 min. Samples were washed with PBS three times and sensitized with the Vectastain ABC kit (PK-6101; Vector Laboratories). After three washes with PBS, color was developed with DAB Peroxidase Substrate Kit (SK-4100; Vector Laboratories), followed by quenching in distilled water. Slides were counterstained with hematoxylin (7211; Thermo Fisher Scientific), dehydrated in xylenes, and coverslips applied with Cytoseal XYL (8312-4; Thermo Fisher Scientific).

Slides were imaged on an Olympus BX51 microscope using Neurolucida software. Quantification was completed in FIJI/ImageJ (Schindelin et al, 2012) by thresholding to select image areas positive for immunostaining. Experimenters were blind to genotype for antibody application, image acquisition, and quantification.

For gross pathology, H&E-stained sections were scored by a veterinary pathologist (Phoenix Central Laboratories) blinded to the genotype for immune cell infiltration, hepatocyte swelling, fibrosis, lipid accumulation, and necrosis. Each category was scored on a four-point scale (0 = absent, 1 = minimal to mild, 2 = moderate, and 3 = severe).

## Plasma profiling

Clinical chemistry plasma analysis for 12 analytes (Table S3) was completed using an Atellica Clinical Chemistry Analyzer (Siemens) at Phoenix Central Laboratories. Analysis for 42 inflammatory markers (Table S5) was completed using a rodent multianalyte panel (Rodent MAP 4.0) based on the Luminex platform at Ampersand Biosciences.

Plasma was collected from mice after a lethal injection of sodium pentobarbital (Fatal Plus; Henry Schein) at 176 d (SD = 2.2 d) or 387 d (SD = 2.1 d). For all analytes (except bile acids), blood was collected via cardiac puncture. Plasma for bile acids was collected via submandibular vein from live mice at 62 ± 11 d. Plasma was flash-frozen after collection in heparinized microtainers (cat. no. 365965; BD) and purification by centrifugation at 1,300$g$ for 10 min, followed by 2,500$g$ for 15 min.

## Statistical analysis

Statistics were processed in R (R Core Team, 2022). For RNA-seq statistics, see the RNA-seq section. For all other data, comparisons with two groups used Welch's $t$ tests to account for unequal variance between groups. To account for multiple comparisons, we adjusted alpha levels using a Bonferroni adjustment. For factorial tests, we used ANVOA with Tukey's tests for post hoc analyses. For summary of tests and group sizes, see Table S6. Data presented in Figs 1, 2, 5–8 used boxplots—horizontal lines indicate 25th, 50th, and 75th percentiles, whereas the vertical whiskers indicate the range of data. Data falling outside 1.5 times the interquartile range are graphed as isolated points but were not excluded from statistical analysis. Graphics were produced using ggplot2 (Wickham, 2016) and Illustrator (Adobe) and BioRender.com.

# Data Availability

All relevant datasets have been uploaded as source data files here and are available at the University of Washington ResearchWorks Repository (uniform resource identifier http://hdl.handle.net/1773/50042).

# Supplementary Information

# Acknowledgements

Funding provided by CHDI Foundation Grant to JB Carroll, NINDS Grant NS124728 to SO Zeitlin, NIDDK Grants R01DK062277 and R01DK103775 to SP Monga, and diagnostic and technical assistance for liver histology was provided by Pittsburgh Liver Research Center's Clinical Biospecimen Repository and Processing Core which is funded by NIDDK Grant P30DK120531 (PI: SP Monga).

# Author Contributions

RM Bragg: conceptualization, data curation, formal analysis, supervision, investigation, visualization, project administration, and writing—original draft, review, and editing.
SR Coffey: conceptualization, data curation, investigation, visualization, project administration, and writing—review and editing.
JP Cantle: data curation, investigation, and writing—review and editing.
S Hu: investigation.
S Singh: investigation.
SRW Legg: investigation.
CA McHugh: data curation and investigation.
A Toor: formal analysis and investigation.
SO Zeitlin: conceptualization, resources, and writing—original draft, review, and editing.

S Kwak: conceptualization, resources, and writing—original draft, review, and editing.

D Howland: conceptualization, resources, and writing—original draft, review, and editing.

TF Vogt: conceptualization, resources, and writing—original draft, review, and editing.

SP Monga: conceptualization, resources, supervision, funding acquisition, methodology, and writing—original draft, review, and editing.

JB Carroll: conceptualization, resources, formal analysis, supervision, funding acquisition, visualization, project administration, and writing—original draft, review, and editing.

## Conflict of Interest Statement

S Kwak, D Howland, and TF Vogt are full-time employees of CHDI Foundation; JB Carroll has consulting income from Cajal Neuroscience and Guidepoint; SP Monga is on the scientific advisory board or a consultant for Antlera, Surrozen, Alnylam, and Vicero Inc., in addition to non-related research funding from Alnylam and Fog Pharmaceuticals.

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
