## [Reviewer comments · Life Science Alliance]

Life Science Alliance

Huntingtin loss in hepatocytes is associated with altered metabolism, adhesion, and liver zonation

Robert Bragg, Sydney Coffey, Jeffrey Cantle, Shikai Hu, Sucha Singh, Samuel Legg, Cassandra McHugh, Amreen Toor, Scott Zeitlin, Seung Kwak, David Howland, Thomas Vogt, Satdarshan Monga, and Jeffrey Carroll

DOI: <https://doi.org/10.26508/lsa.202302098>

Corresponding author(s): Jeffrey Carroll, University of Washington

Review Timeline:

Submission Date:	2023-04-15
Editorial Decision:	2023-05-15
Revision Received:	2023-07-17
Editorial Decision:	2023-08-18
Revision Received:	2023-08-24
Accepted:	2023-08-28

Transaction Report:

May 15, 2023

Re: Life Science Alliance manuscript #LSA-2023-02098-T

Jeffrey Bryan B Carroll
University of Washington
Department of Neurology

Dear Dr. Carroll,

Thank you for submitting your manuscript entitled "Huntingtin loss in hepatocyte leads to impaired beta-catenin signaling and aberrant liver zonation" to Life Science Alliance. The manuscript was assessed by expert reviewers, whose comments are appended to this letter. We invite you to submit a revised manuscript addressing the Reviewer comments.

Thank you for this interesting contribution to Life Science Alliance. We are looking forward to receiving your revised manuscript.

Sincerely,

B. MANUSCRIPT ORGANIZATION AND FORMATTING:

Reviewer #1 (Comments to the Authors (Required)):

1. The manuscript describes a novel mouse model generated by the Authors, which is the hepatocytes-specific knockout of HTT. The title of the manuscript is "Huntingtin loss in hepatocytes leads to impaired beta-catenin signaling and aberrant liver zonation in mice". To justify the conclusion in the title, the authors use mainly RNAseq transcriptional methods and extensive analysis of the data supported by a variety of statistical methods. The Authors identify altered expression of genes representing the zones of the liver (actually zones of the liver lobules) in knockout mice. The Authors demonstrate that the pericentral zone genes are downregulated while the periportal zone genes have no clear direction of change. There are also some basic histological tests that show the microscopic structure of liver lobules and metabolic tests of the mice.

2. The title of the manuscript does not mention that the main tool used in work is transcriptomics and that the analysis is done in hepatocyte-specific knockout. Instead, the title contains words like signaling and zonation, which is clearly associated with the use of methods that investigate the transduction and signaling pathways such as p-Ab, pulldown of proteins, siRNA knockdown, or overexpression of signaling proteins or intensive histological testing of lobule zonation, staining or co-immunostainings for zone markers, sorting cells on cell sorter and RNAseq with these populations characteristic to zones or perhaps scRNAseq to find out how the zonation in the absence of HTT exactly looks like. For concluding signaling, it is, for instance, not sufficient to say that the mice with β -catenin knockout in hepatocytes show increased basal bile acid levels similar to HttLKO/LKO to justify the altered β -catenin signaling.

So the right title at the present stage of the work that would reflect the content should be, for instance:

Liver metabolic phenotype and transcriptional signatures of the lobular zones are changed in hepatocyte-selective HTT knockout.

Similarly, the conclusions in the manuscript should be carefully considered and changed, and otherwise, to justify the present title and conclusions, the work needs at least some selection of new experiments mentioned above, which will be time-consuming.

3. I am not sure how the acetaminophen challenge experiment fits the rest of the manuscript in terms of the message. It wakes more mechanistic questions and starts a new branch

4. The manuscript also contains a good portion of speculation on the developmental origin of the phenomena seen in hepatocyte knock-out mice. This is very interesting and probably true however not quite have support in experiments done. For instance, the Authors used 10 months old mice for RNAseq, but to look at least in part at developmental aspects would need RNAseq or looking at liver microstructure of young mice.

5. The statistics in the manuscript may be a minor issue, but it turns out to be disturbing and dismisses the reader's focus. The description of the statistics in the dedicated section is modest, while the figure legend is packed with the statistical parameters, and I have doubts that it is always needed, and if it is needed, it can be summarized in a supplementary table (excel file). Moreover, the parameters are not always clearly indicated. I do not know what B is in fig1. Is it Bonferroni? but it is, in fact, a t-test, so why not "t"? Is the listing of the t value and F value of absolute necessity in the figure and results text? Would t-test and p-value suffice? Or if authors have ANOVA, then put the p-value for ANOVA and the p-value of Turkey. And also, what is the (x,y) with t and F? Are these the degrees of freedom everywhere? Are they of absolute necessity to put in text and legend? Ultimately, the most important for the reader is: what and how much it is changed and if it is significant (and mechanistically why). In addition, what exactly does that mean? : (62/1019, odds ratio = 1.77, p = 0.02) What was the exact message to the reader?

6. The text. For instance, the introduction can benefit from more details and removing some redundancy. The first and second paragraphs are devoted to HD and HTT, and in the introduction, there are no clear points, assumptions, or hypotheses. Why exactly is genetic fitness discussed in the second paragraph? The section would benefit from describing the liver lobular structure and genes or metabolism characteristics for zones.

Reviewer #2 (Comments to the Authors (Required)):

The manuscript entitled "Huntingtin loss in hepatocytes leads to impaired beta-catenin signaling and aberrant liver zonation in mice" discusses the adverse effect of loss of Huntington expression on liver function and has established that this loss is accompanied by a loss of pericentral zone to peri pleural zone. The paper overall seems promising, and I have a few comments to discuss.

1. No clear rationale of studying HTT in the liver.
2. Fig. 1(B-C), why is the relative HTT level more in lko mice than in wild types?
3. ...and are indistinguishable from littermates in terms of gross appearance and behavior....However, there is significant weight difference between the two (Fig.1 (D))
4. There is a significant difference in the CDH1 data from IHC corresponding RNA seq transcript levels; how would authors explain this?
5. Resistance to toxic doses of acetaminophen in httLKO has been related to impaired catenin signaling seems too theoretical.
6. No clear conclusion.
7. Only female mice have been used for the study.
8. Why have n=300 mice been used for Figure 3A, despite all the Ns being positive? Please clarify.
9. The quality of figures and presentation of data can be improved.
10. HTT KO impairs liver function and zonation but protects from Acet-induced liver damage.

Reviewer #3 (Comments to the Authors (Required)):

Study by Bragg et al describes the phenotype of HTT hepatocyte specific knockout mice. Authors demonstrate multiple physiological changes, including changes in bile acids, cholesterol and urea, hypoglycemia, and impaired adhesion in knockout mice that mimic loss of β -catenin activity. This study has important implications since HTT reduction for treatment of Huntington's disease would have potential off target effects on other organs such as liver.

Comments:

1. Authors present data that suggest that β -catenin signaling is impaired in Htt KO mice, however β -catenin activity/expression/localization etc were not examined.
2. Htt KO mice seem to have reduced levels of Wnt2 and Wnt9b resulting in reduced pericentral specific gene expression. It is not clear how hepatocyte HTT regulate gene expression in endothelial cells. This is intriguing finding and needs more discussion of the potential mechanism involved in this dysregulation.
3. Authors report that cell adhesion was impaired in KO mice. However, adhesion pathways were not enriched in RNA-seq data analysis. What is the mechanism of adhesion dysfunction in KO mice?
4. Authors present data from mice of different ages. It would be helpful to summarize the phenotypes at different ages, which changes come first, and which ones develop later.
5. Sex of the experimental animals is not clear. Were only female mice used for all experiments?

Reviewer #1 (Comments to the Authors (Required)):

We thank the reviewer for their careful reading of our manuscript, which they accurately summarize below. Our responses to each of the concerns they raised are below in bold, interspersed with their original comments.

1. The manuscript describes a novel mouse model generated by the Authors, which is the hepatocytes-specific knockout of HTT. The title of the manuscript is "Huntingtin loss in hepatocytes leads to impaired beta-catenin signaling and aberrant liver zonation in mice". To justify the conclusion in the title, the authors use mainly RNAseq transcriptional methods and extensive analysis of the data supported by a variety of statistical methods. The Authors identify altered expression of genes representing the zones of the liver (actually zones of the liver lobules) in knockout mice. The Authors demonstrate that the pericentral zone genes are downregulated while the periportal zone genes have no clear direction of change. There are also some basic histological tests that show the microscopic structure of liver lobules and metabolic tests of the mice.

2. The title of the manuscript does not mention that the main tool used in work is transcriptomics and that the analysis is done in hepatocyte-specific knockout. Instead, the title contains words like signaling and zonation, which is clearly associated with the use of methods that investigate the transduction and signaling pathways such as p-Ab, pulldown of proteins, siRNA knockdown, or overexpression of signaling proteins or intensive histological testing of lobule zonation, staining or co-immunostainings for zone markers, sorting cells on cell sorter and RNAseq with these populations characteristic to zones or perhaps scRNAseq to find out how the zonation in the absence of HTT exactly looks like. For concluding signaling, it is, for instance, not sufficient to say that the mice with β -catenin knockout in hepatocytes show increased basal bile acid levels similar to HttLKO/LKO to justify the altered β -catenin signaling.

So the right title at the present stage of the work that would reflect the content should be, for instance: Liver metabolic phenotype and transcriptional signatures of the lobular zones are changed in hepatocyte-selective HTT knockout. Similarly, the conclusions in the manuscript should be carefully considered and changed, and otherwise, to justify the present title and conclusions, the work needs at least some selection of new experiments mentioned above, which will be time-consuming.

Response: We thank the reviewer for these comments, and agree that the title could more precisely describe our actual data. As a result, we have changed the original title: "Huntingtin loss in hepatocytes leads to impaired beta-catenin signaling and aberrant liver zonation in mice" to "Huntingtin loss in hepatocytes is associated with altered metabolism, adhesion and liver zonation". After some discussion about alternatives, we feel that "liver zonation" is accurate, and reflective of the language used in the literature around this topic (e.g. PMID: 36220068, 31535084, 34924168). However, if the reviewers and editor feel strongly we are happy to reconsider.

3. I am not sure how the acetaminophen challenge experiment fits the rest of the manuscript in terms of the message. It wakes more mechanistic questions and starts a new branch

Response: This was, to us, an important validation that the transcriptional and histological changes we observe in the $Htt^{LKO/LKO}$ mice have physiological relevance. We were interested in demonstrating that the observed gene expression changes, particularly altered expression patterns of key cytochrome P450 genes, are associated with functional outcomes at the physiological level. Previous studies (PMID: 16557553, 19679878, 24700412) had shown that reduced expression of these enzymes results in reduced conversion of acetaminophen to a toxic metabolite (N-acetyl-p-benzo-quinone imine, NAPQI). Indeed, we showed here that the $Htt^{LKO/LKO}$ mice have a striking resistance to acetaminophen induced toxicity, which mimics that of liver-specific knockout mice, hopefully anchoring our study within the broader literature in this area. While we don't think that this phenotype is important in and of itself, we think it suggests that the histological and transcriptional changes impact physiology.

4. The manuscript also contains a good portion of speculation on the developmental origin of the phenomena seen in hepatocyte knock-out mice. This is very interesting and probably true however not quite have support in experiments done. For instance, the Authors used 10 months old mice for RNAseq, but to look at least in part at developmental aspects would need RNAseq or looking at liver microstructure of young mice.

Response: We may have been overly cautious in how we described our concern about the potential developmental confounds of our system. We have tried to clarify in this version of the discussion that we are raising the developmental loss of HTT in hepatocytes simply as a potential confound of our experimental system, and not to speculate too much on the consequences of this confound.

5. The statistics in the manuscript may be a minor issue, but it turns out to be disturbing and dismisses the reader's focus. The description of the statistics in the dedicated section is modest, while the figure legend is packed with the statistical parameters, and I have doubts that it is always needed, and if it is needed, it can be summarized in a supplementary table (excel file). Moreover, the parameters are not always clearly indicated. I do not know what B is in fig1. Is it Bonferroni? but it is, in fact, a t-test, so why not "t"?

Response: In our initial description of our gene expression studies we have reported a B-statistic from RNAseq analysis, which is the log of the likelihood that a gene is differentially expressed (Ritchie et al, 2015). To make things more clear, we only report the adjusted p value to remove distraction caused by specialized statistical measures. Throughout, as outlined below, we have tried to simplify and streamline the presentation of the statistical results in this version of the draft.

Is the listing of the t value and F value of absolute necessity in the figure and results text? Would t-test and p-value suffice? Or if authors have ANOVA, then put the p-value for ANOVA and the p-value of Turkey. And also, what is the (x.y) with t and F? Are these the degrees of freedom everywhere? Are they of absolute necessity to put in text and legend? Ultimately, the most important for the reader is: what and how much it is changed and if it is significant (and mechanistically why). In addition, what exactly does that mean? : (62/1019, odds ratio = 1.77, p = 0.02) What was the exact message to the reader?

Response: To simplify things for the reader and make the conclusions easier to understand, we will do the following: we will remove and simplify the statistics so they are present in the figure legends or text only. With regards to the degrees of freedom, after some discussion we have decided to leave them in this version as it aids the reader in quickly understanding the sample sizes for each comparison. With regards to the EnrichR pathways, we have clarified that 62/1019 genes are present, but we feel that the odds ratio (OR) is a common metric so we will leave it, with an additional reference to the statistical basis for EnrichR OR test in the RNA sequencing methods section.

6. The text. For instance, the introduction can benefit from more details and removing some redundancy. The first and second paragraphs are devoted to HD and HTT, and in the introduction, there are no clear points, assumptions, or hypotheses. Why exactly is genetic fitness discussed in the second paragraph? The section would benefit from describing the liver lobular structure and genes or metabolism characteristics for zones.

Response: We thank the reviewer for these suggestions and have modified the introduction text in response with an eye towards clarity.

Reviewer #2 (Comments to the Authors (Required)):

We thank the reviewer for reviewing our manuscript, and for their thoughtful comments below. We have addressed each of these below, in bold, interspersed with all of their original comments.

The manuscript entitled "Huntingtin loss in hepatocytes leads to impaired beta-catenin signaling and aberrant liver zonation in mice" discusses the adverse effect of loss of Huntington expression on liver function and has established that this loss is accompanied by a loss of pericentral zone to peri pleural zone. The paper overall seems promising, and I have a few comments to discuss.

1. No clear rationale of studying HTT in the liver.

Response: We apologize and agree with the reviewer that we did not explain our rationale well in the introduction. We have added that text to the current version.

2. Fig. 1(B-C), why is the relative HTT level more in lko mice than in wild types?

Response: Although the quantified blot levels look slightly higher, this difference is not statistically significant.

3. ...and are indistinguishable from littermates in terms of gross appearance and behavior....However, there is significant weight difference between the two (Fig.1 (D)).

Response: Body weights are lower, however this is not noticeable by eye and the gross appearance is the same.

4. There is a significant difference in the CDH1 data from IHC corresponding RNA seq transcript levels; how would authors explain this?

Response: We believe there is a qualitative difference in the staining pattern of CDH1. Our quantification method of measuring 'total area positive' for CHD1 immunoreactivity was likely not sensitive enough to pick up on CDH1 reduction that is predicted by the reduction of *Cdh1* transcripts. We are planning future studies to better understand the altered zonation of transcripts and protein in these mice and how one readout (e.g., transcripts) relates to another (e.g., protein).

5. Resistance to toxic doses of acetaminophen in httLKO has been related to impaired catenin signaling seems too theoretical.

Response: We agree with the reviewer that this is a surprising link. However, the link between beta-catenin loss in hepatocytes and altered gene expression - including reductions in key cytochrome P450 genes - has been used in multiple published studies (e.g. PMID6557553, 19679878, 24700412) as a functional validation of the impact of altered beta-catenin signaling on hepatic physiology. We felt that this experiment helped ground the study in the broader

literature around the link between beta-catenin signaling and altered CYP450 gene expression.

6. No clear conclusion.

Response: We apologize for leaving some of our conclusions and intentions with the study a little vague in our original version of this manuscript. In the updated draft we have attempted to adjust the language in the introduction and discussion to more precisely frame the motivations and conclusions of the study.

7. Only female mice have been used for the study.

Response: Due to constraints of *in vivo* mouse work, some of our experiments only used female mice in order to group-house non litter-mates to alleviate space constraints in our mouse colony. Where possible, we also used mixed-sex cohorts. We have added a supplemental table (Table S6) that outlines the sex used for each experiment to help clarify this.

8. Why have n=300 mice been used for Figure 3A, despite all the Ns being positive? Please clarify.

Response: We reported large Ns to highlight the clear consistency with which we see this phenotype. We regularly perfuse livers to extract primary hepatocytes for other downstream use and thus have very high observed Ns for this particular phenotype that we have not previously reported.

9. The quality of figures and presentation of data can be improved.

Response: We have provided the journal with higher quality versions of the figures and have simplified the figure captions in order to make the presentation of the data more clear.

10. HTT KO impairs liver function and zonation but protects from Acet-induced liver damage.

Response: Indeed, liver function and zonation is 'impaired' which actually protects these mice from acetaminophen induced liver damage. It is possible that the impaired function and zonation also creates liabilities, rather than protection, to other xenobiotic stressors. We will be assessing many xenobiotic stressors with clinical relevance in future studies.

Reviewer #3 (Comments to the Authors (Required)):

We thank the reviewer for taking the time to review our manuscript and provide feedback. We have addressed each of their points below, in bold, interspersed with all of their original comments.

Study by Bragg et al describes the phenotype of HTT hepatocyte specific knockout mice. Authors demonstrate multiple physiological changes, including changes in bile acids, cholesterol and urea, hypoglycemia, and impaired adhesion in knockout mice that mimic loss of β -catenin activity. This study has important implications since HTT reduction for treatment of Huntington's disease would have potential off target effects on other organs such as liver.

Comments:

1. Authors present data that suggest that β -catenin signaling is impaired in Htt KO mice, however β -catenin activity/expression/localization etc were not examined.

Response: We think that β -catenin impairment may be due to the loss of HTT which acts as a scaffold for some B-catenin complexes. We will examine this specific interaction in future studies.

2. Htt KO mice seem to have reduced levels of Wnt2 and Wnt9b resulting in reduced pericentral specific gene expression. It is not clear how hepatocyte HTT regulate gene expression in endothelial cells. This is intriguing finding and needs more discussion of the potential mechanism involved in this dysregulation.

Response: We thank the reviewer for this comment and astute observation. In fact, we think it is potentially amongst the most important findings of our paper, as it suggests a potentially non-cell-autonomous relationship between HTT levels in hepatocytes and Wnt gene expression levels in adjacent endothelial cells, whose HTT levels we do not predict to be reduced in our hepatocyte HTT knockout mice. Our lab is currently working on large-scale single cell analyses in follow-up studies. We reasoned that this first descriptive manuscript set the stage for follow-up studies into the mechanisms by which HTT loss in one cell type potentially influences gene expression of Wnts in adjacent cell types. In our preliminary ongoing experiments we have replicated the finding that adult-onset HTT lowering (via a different mechanism) leads to Wnt2/9b reductions in the liver, suggesting that the data in this paper are robust across different HTT lowering contexts.

3. Authors report that cell adhesion was impaired in KO mice. However, adhesion pathways were not enriched in RNA-seq data analysis. What is the mechanism of adhesion dysfunction in KO mice?

Response: There were some cell adhesion pathways that were enriched in our RNAseq data set (For example, positive regulation of cell-cell adhesion mediated by integrin - GO:0033634, Fig 4D). The mechanism of adhesion dysfunction, however the mechanism leading to the adhesion phenotype seen here is unknown. We are currently following up with studies to understand this mechanism, including IHC of several adhesion proteins and EM of cell-cell junctions.

4. Authors present data from mice of different ages. It would be helpful to summarize the phenotypes at different ages, which changes come first, and which ones develop later.

Response: Due to constraints of *in vivo* mouse work, many of our experiments occurred at different ages. We ensured that experiments all occurred with adult mice (6 months or greater) to examine the phenotypes after development. We have added a supplemental table (Table S6) that outlines the age, sex, and statistical test used for each experiment to help understand the age each phenotype was observed.

5. Sex of the experimental animals is not clear. Were only female mice used for all experiments?

Response: Due to constraints of *in vivo* mouse work, some of our experiments only used female mice in order to group-house non-littermates to alleviate space constraints in our mouse colony. Where possible, we also used mixed-sex cohorts. We clarified this in the text and added a supplemental file (Table S6) that outlines the sex, age, and statistical test used for each experiment.

August 18, 2023

RE: Life Science Alliance Manuscript #LSA-2023-02098-TR

Dr. Jeffrey B Carroll
University of Washington
Department of Neurology
HMC #359660
325 9th Ave
Seattle, WA 98104-2499

Dear Dr. Carroll,

Thank you for submitting your revised manuscript entitled "Huntingtin loss in hepatocytes is associated with altered metabolism, adhesion, and liver zonation". We would be happy to publish your paper in Life Science Alliance pending final revisions necessary to meet our formatting guidelines.

- please add a Summary Blurb/Alternate Abstract in our system
- please consult our manuscript preparation guidelines <https://www.life-science-alliance.org/manuscript-prep> and make sure your manuscript sections are in the correct order
- please add the Twitter handle of your host institute/organization as well as your own or/and one of the authors in our system
- please add your main, supplementary figure, and table legends to the main manuscript text after the references section
- please add a callout for Figure 3C, Figure 5B, Figure 7B, Figure 8A, Figure 8B to your main manuscript text

A. FINAL FILES:

B. MANUSCRIPT ORGANIZATION AND FORMATTING:

Sincerely,

Reviewer #1 (Comments to the Authors (Required)):

I am satisfied with the responses and have no further comments.

Reviewer #2 (Comments to the Authors (Required)):

3. ...and are indistinguishable from littermates in terms of gross appearance and behavior....However, there is significant weight difference between the two (Fig.1 (D)).

Response: Body weights are lower, however, this is not noticeable by eye and the gross appearance is the same.

Something that is significantly lower by statistics but is not noticeable to your eye is considered to be not significant.

I am not going to read the rest of the rebuttal letter, sorry.

August 28, 2023

RE: Life Science Alliance Manuscript #LSA-2023-02098-TRR

Dr. Jeffrey B Carroll
University of Washington
Department of Neurology
HMC #359660
325 9th Ave
Seattle, WA 98104-2499

Dear Dr. Carroll,

Thank you for submitting your Research Article entitled "Huntingtin loss in hepatocytes is associated with altered metabolism, adhesion, and liver zonation". It is a pleasure to let you know that your manuscript is now accepted for publication in Life Science Alliance. Congratulations on this interesting work.

DISTRIBUTION OF MATERIALS:

Again, congratulations on a very nice paper. I hope you found the review process to be constructive and are pleased with how the manuscript was handled editorially. We look forward to future exciting submissions from your lab.

Sincerely,
